# Morphology, ultrastructure and function of the sternal gland in two mason bee species (*Osmia bicornis* and *O. cornuta*)

Stephanie Krüger[1], Karsten Seidelmann[2]*

1 Core Facility Microscopy, Biozentrum, Martin-Luther-University Halle-Wittenberg, Halle (Saale), Germany, 2 Department Animal Physiology, Institute of Biology/Zoology, Martin-Luther-University Halle-Wittenberg, Halle (Saale), Germany

* karsten.seidelmann@zoologie.uni-halle.de

## Abstract

Pheromonal communication is often facilitated by the evolution of specialized gland structures. Males of two species of mason bees release specific carboxylic esters from a hidden sternal gland. This gland tissue consists of a single layer of class 1 secretory cells located on two sternites. The unmodified cuticle above the gland tissues is dented and covered with setae. Together with the elongated covering sternite, these depressions form a storage vessel for volatiles. This gland morphology facilitates the slow, continuous release of volatiles through leakage from the cavity, as well as the enhanced, voluntary release of higher concentrated pheromone puffs through abdominal movements. The carboxylic esters released from the gland have a deterrent effect and indicate the sex of the male from a distance. Pheromone puffs emitted by abdominal movements enable mating to occur without interference from competitors. In this context, the carboxylic esters act as antiaphrodisiacs.

## Introduction

The reproductive investment of males in mason bees is limited to the insemination of females. Therefore, finding a receptive female and mating with her is the central task of the male's entire life. This task is complicated by the distinct male-based sex ratio and by the monandry of females, who are receptive immediately upon emerging from their maternal nest [1,2]. Protandry is therefore a common strategy to avoid missing mating opportunities [3]. This scenario also applies to two widespread solitary mason bees, the red mason bee (*Osmia bicornis* Linnaeus, syn. *O. rufa* L.) and the horn-faced mason bee (*O. cornuta* Latreille). Males of both species are forced into a scramble competition polygyny for mating [4]. They patrol nest sites and flowers, pouncing on any object resembling a female present at a rendezvous site [4,5]. However, males rely on volatiles to identify a potential mate [6,7]. Females of both mason bee species do not release specific sex pheromones [8] but they can be identified by their nuptial (virgin) cuticular hydrocarbon

**Data availability statement:** The data and raw immages are available from the repository of the University Halle ("Share_it - Open Access und Forschungsdaten-Repositorium der Hochschulbibliotheken in Sachsen-Anhalt"). URL: https://opendata.uni-halle.de//handle/1981185920/122632 DOI: http://dx.doi.org/10.25673/120677.

**Funding:** The author(s) received no specific funding for this work.

**Competing interests:** The authors have declared that no competing interests exist.

(CHC) bouquet [9]. This CHC bouquet is modified within the very first days after emergence [8]. *O. bicornis* males are tuned to the nuptial bouquet and are able to discriminate between receptive and non-receptive females [6,9]. However, freshly emerged males of both *O. cornuta* and *O. bicornis* share the same respective CHC pattern with conspecific newly emerged virgin females [10]. They can easily be mistaken for receptive females and are occasionally engaged in same-sex sexual behaviour (SSB) by older males [10]. While males alter the CHC bouquet within the first few days to prevent further sexual harassment, they additionally start releasing large quantities of species-specific carboxylic acid esters within three days of emerging. These esters dominate the volatile bouquet after a few days [10]. *O. bicornis* males release ethyl (Z)-7-hexadecenoate (7-C16:1-EE) [9,11], while *O. cornuta* males produce a mixture of isopropyl-9-hexadecenoate (9-C16:1-IPE) and ethyl-9-hexadecenoate (9-C16:1-EE) [10].

The source of the sex-specific esters has been identified in both species of mason bees studied here as a sternal gland located on the abdomen of male bees [9,10]. Sternal glands are a specific type of exocrine epidermal glands found on the sternites (ventral segments) of the abdomen in various taxa, and they are especially widespread among social species (for review: [12]; Heteroptera: [13,14]; Isoptera: [15,16]; Hymenoptera: ants [17,18] and wasps [19–23]; Thysanoptera: [24,25]). The term "sternal gland" has not always been used consistently in the literature [23,26]. In some cases, external cuticular structures such as pore plates have also been referred to as sternal glands, even though they represent only part of the overall glandular system [26].

Beside this variation in terminology, there is also a great diversity in the size, morphology, chemistry, release mechanisms, and functions of exocrine glands across insects in general [27]. They secrete trail pheromones primarily in ants [28,29] and termites [16], defence substances in polistine wasps [30] and recruiting pheromones in hornets [31]. However, the function of the male-specific esters produced by the sternal glands of mason bees has not yet been fully clarified. In *O. bicornis,* the pheromone was originally considered to act as an antiaphrodisiac applied at the post-copulatory display [9,32], rendering inseminated females unattractive to other males [9,11]. However, this function has been called into question, since the pheromone cannot be detected on females immediately after copulation [8]. The observation that only males that have recently emerged and lack the pheromone become victims of SSB in both mason bee species, as well as the coincidence of the change in the eclosion CHC bouquet with the onset of ester release, supports the alternative hypothesis that the esters act as deterrent pheromones that extend the range of their sex-specific CHC scent tag (self-marking antiaphrodisiac hypothesis) [10].

The aim of this study is to provide a detailed description of the morphology and histology of the sternal gland in both mason bee species. Based on this anatomical foundation, behavioural assays will be employed to test the self-marking antiaphrodisiac hypothesis in both species. Using SSB between males, we assessed whether newly emerged males treated with gland extracts containing male-specific esters are less attractive to other males searching for mates. Furthermore, given the somewhat similar pheromone mixtures of the two sympatric species, we also explored the possibility of interspecific signalling.

## Materials and methods

### Study bees

The bees used in this study were obtained from regularly maintained populations of *O. bicornis* and *O. cornuta* located at the Botanical Garden of Halle/Saale (Germany, Saxony-Anhalt, 51° 29' 04'' N, 11° 56' 07'' E; permit RL-0590, Landesamt für Umweltschutz Sachsen-Anhalt). Individuals were reared from cocoons stored at 4°C after hibernation in the laboratory. To allow emergence, male cocoons were transferred to emergency boxes and kept at room temperature (21–22°C) with natural daylight. Bees emerged preferentially in the morning hours [33] and were immediately used for bioassays. Bees for sternal gland morphology and gland extracts were kept in plastic boxes and fed on a honey/water solution (2/1) until sampling. Prior to dissection of males for histology or gland extraction, the animals were cooled down and decapitated or freeze-killed.

### Light microscopy and transmission electron microscopy

Sternites were dissected using fine tweezers and immediately fixed in 3% glutaraldehyde in 0.1 M Na-Cacodylate buffer (pH 7.4, 3h). Postfixation (1h in 1% OSO4 in buffer) and en-bloc staining (1h in 70% uranyl acetate in 70% ethanol) were performed during stepwise dehydration with ethanol. Infiltration and embedding were carried out in SPURR's resin. Blocks were sectioned using a Leica Ultramicrotome S to produce semithin sections (500 nm, light microscopy) and ultrathin sections (70 nm, TEM) on glass slides and copper grids, respectively. Semithin sections were stained with 0.1% toluidine blue in 2.5% NaHCO3 for 15 min and covered with Euparal. In order to examine the cuticular structure of the sternites, whole mount samples were produced. To make the cuticle transparent, the sternites were lightened by 5% NaOH overnight at 45°C, followed by stepwise dehydration in ethanol, clearing with terpineol (5 min) and embedding in Euparal.

Imaging was performed using a Leica M165C equipped with Planapo 0.63, Flexicam C3, and capture software Leica LASX V. 5.0.3, and a Leica DMBR equipped with a Leica DFC450 camera. Ultrathin sections were stained with uranyl acetate and lead citrate using a Leica AC20 and imaged with a Zeiss EM900 at 80kV equipped with a water-cooled 1k slow scan CCD camera (TRS-Tröndle Restlichtverstärkersysteme). Image post-processing was conducted using Adobe Photoshop CS6 (brightness-, contrast-, tone correction, stitching, high-pass filter as needed). Illustration was created with BioRender (https://BioRender.com).

### Scanning electron microscopy

Scanning electron microscopy was performed on whole metasomas and dissected sternites and macerated sternite whole mounts. Whole metasomas were separated from decapitated animals and a window was cut in the top right corner to facilitate better penetration of the fixative. Samples were fixed in Carnoy (ethanol: chloroform: glacial acetic acid 6:3:1) for 2h. After dehydration, metasomas were embedded in paraffin via isopropanol. Hardened paraffin blocks were sectioned on a rotational microtome until the target region was reached. Samples were then deparaffinized with Roticlear, washed in ethanol and acetone, and critical point dried (Emitech K850). Sternite samples were handled identically, but without paraffin embedding. To reveal the chitin surface, the adhering glandular tissue and wax deposits were digested by Enzyrim OSA (Subtilisin, Bauer Handels GmbH, Switzerland) overnight at 60°C, stepwise dehydration and chemical drying with HMDS. Chemical and Critical Point dried samples were glued on aluminium stub holders. Macerated sternite samples were sputtered with 30 nm chromium as needed and imaged with a Zeiss Gemini 500 at 0.5–2 kV.

### Behavioural assay

The effect of gland extractions on the SSB of mate seeking mason bee males was tested in the field. Early in the regular flight season, at the time of male emergence, freshly emerged males were presented to males searching for a virgin female in front of artificial nest aggregations where plenty of males were searching for emerging females. Field

experiments with free-flying males under natural conditions were used to ensure normal motivations and behaviour of searching males and to exclude any laboratory artefacts or behavioural modifications due to cage conditions. However, a drawback of field conditions, was the changing number and activity of males searching for newly emerged females at a nest aggregation between the test days, which was also affected by the weather conditions. The test males were incubated in the laboratory at room temperature. Bees emerged preferentially in the morning hours [33]. Only freshly emerged males without pheromone [10] were used for the tests. The males were cooled and tethered to a plastic Petri dish with a fresh piece of thin sewing thread wrapped loosely around their petioles. This allowed them to adopt their normal resting positions and movements while preventing them from walking or flying away. No differences in the response to approaching males were detectable between resting and tethered males. One microlitre of sternal gland extract in pentane, or pentane alone as a solvent control, was slowly applied to the thorax of the tethered males using a microlitre syringe. The Petri dishes with the tethered males were placed directly in front of a nest box or shelter housing several hundred nests of the respective species, where numerous males were seeking for females to emerge. The males were tested only once and then released. After each trial, the Petri dishes were cleaned with an ethanol-soaked paper towel to remove potential contamination. To statistically control for the different conditions on the experimental days (number and motivation of males, weather conditions, etc.), the day of the experiment was included as a fixed factor in the analysis and is referred to as 'trial'.

Male SSB is well pronounced in *O. cornuta*. Newly emerged males are highly attractive to older males and were regularly mounted with several copulation attempts per approach (see results). Therefore, the attractiveness of untreated test males was initially monitored for about 5 min before the application of a test substance (ester or solvent). The gland extracts or the solvent were then applied and the attractiveness was measured again for 5 min. Although *O. cornuta* males were highly attractive to conspecifics, their attractiveness decreased over time, resulting in fewer approaches by the mate-seeking males. This effect became apparent at the time of data analysis and could not be controlled for by adjusting the experimental design. Consequently, parameters quantifying male attractiveness had to be statistically analysed separately before and after application in a repeated measures design, rather than their effect over time within treatment groups. In *O. bicornis*, SSB is less pronounced compared to *O. cornuta*. The attractiveness of the test males to the mate-seeking males declined rapidly, making it impossible to reliably measure a change in the attractiveness of an individual male. Therefore, in *O. bicornis*, test substances were applied directly to newly emerged males without first measuring the untreated attractiveness of individuals.

The attractiveness of test males to mate seeking males was rated by several parameters that were recorded during the 5 min of observation period: mountings with copulation attempts and the number of copulation attempts, number of approaches (flights towards the test male, landing on or next to the test male with physical contact), inspections (darting towards the test male without landing on the male with a short contact "on the wing"). As the time appeared to be critical in these experiments, the attractiveness was monitored for a limited period of time (5 min) rather than a specific number of approaches. Based on these rating parameters, relative parameters such as the number of attempts to copulate per approach and the proportion of physical contacts made on all detectable approaches including inspections had to be calculated to control for the different number and activity of searching males on different experimental days. Finally, the efficacy of the sternal gland extracts was also tested episodically on the other mason bee species. A small sample of males was treated with the sternal gland extract of the other species to check for an effect as a last trial in the natural hatching period of males.

## Results

### Sternal gland morphology

*O. bicornis* and *O. cornuta* possess morphologically similar hidden sternal glands on the ventral side of the metasoma, the posterior part of the abdomen. In the order Apocrita, the propodeum (the first part of the abdomen) is fused to the thorax

and separated from the metasoma by a constriction ('wasp waist'). While we follow the morphological circumstances when counting the sternites, we use the term 'abdomen' instead of 'metasoma' for simplicity. The sternal gland is located beneath the IV$^{th}$ and V$^{th}$ sternites and shielded by the elongated III$^{rd}$ sternite (Fig 1A-1D). The extents of the gland parts are well recognizable by depressions of the sternites bordered by a scalloped edge forming well-defined cavities filled with characteristic hair brushes of long setae (Fig 1C-1F). These hair brushes are more pronounced on the IV$^{th}$ than on the V$^{th}$ sternite (Fig 1C-1F). The setae forming the brushes are curved backwards and point to a distinct hair bristle located in an indented area of the caudal edge of the IV$^{th}$ sternite (Fig 1C, 1E). The part of the gland hosted by the V$^{th}$ sternite is restricted to the area covered by the IV$^{th}$ sternite and the setae are oriented toward the hair bristle of the IV$^{th}$ sternite (Fig 1C-1F). The depressions of the IV$^{th}$ and V$^{th}$ sternite covered by the elongated III$^{rd}$ sternite form setae-filled air caverns (Fig 1G).

The gland tissue itself consists of a monolayer of epithelial gland cells that lie directly beneath the cuticle (Fig 1H). Basally, the cells are bounded by a basal lamina (Fig 2A). Neighbouring cells are connected by zonula adherens, followed by pleated septate junctions (Fig 2B). The gland cells are characterized by a large number of mitochondria, an expanded Golgi apparatus, large basal nuclei, numerous free ribosomes, and vesicles filled with dense material (Fig 2A, 2B).

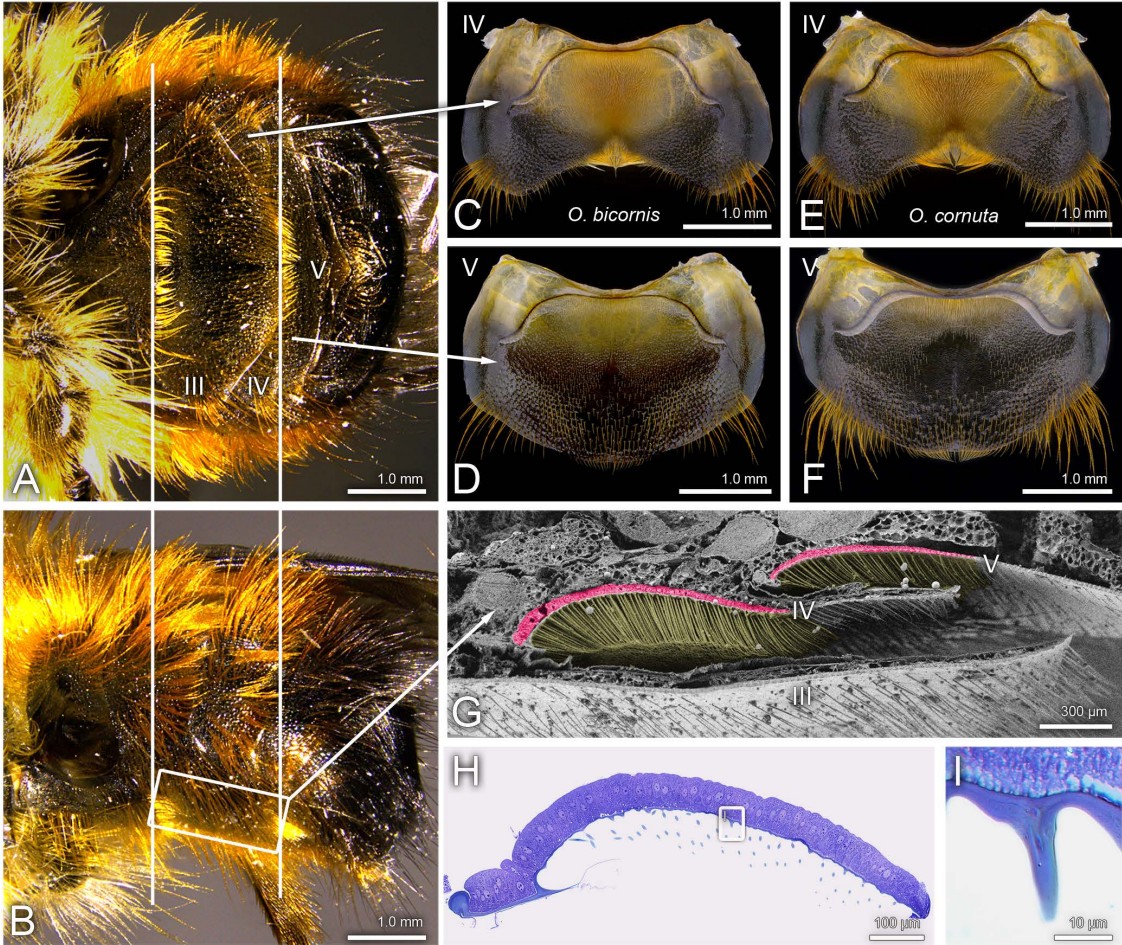

**Fig 1. Localization and morphology of the sternal gland.** A ventral view and B lateral view on the abdomen of *O. bicornis*; C, D morphology of the IV$^{th}$ and V$^{th}$ sternites of *O. bicornis* and E, F of *O. cornuta*; G coloured transection (SEM) through the abdomen in the region of the sternal gland (*O. bicornis*), gland tissue in red, setae in yellow; H sagittal semi-thin transection through the IV$^{th}$ sternite (*O. bicornis*); I detail of truncated setae; numbers indicate the sternites.

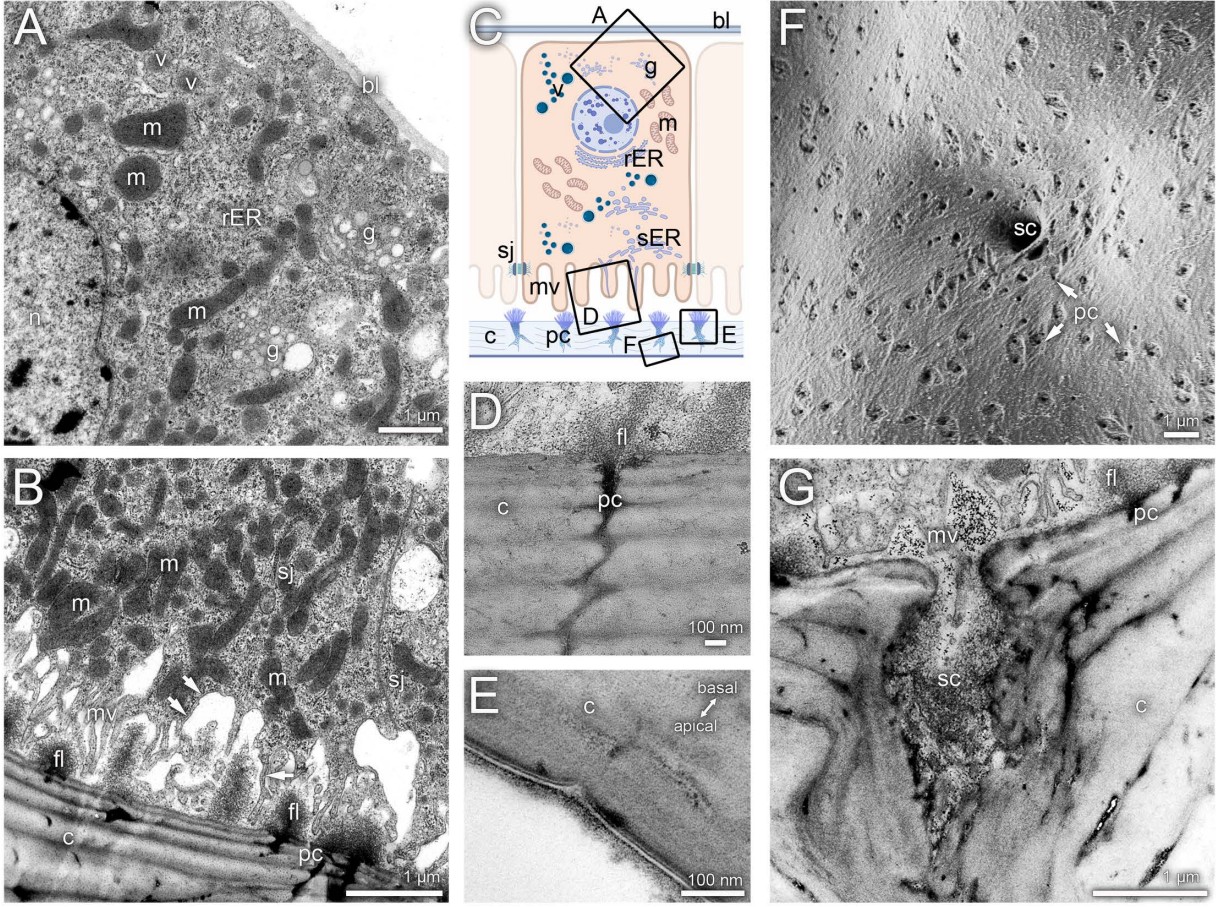

**Fig 2. Histology of sternal gland cells.** A basal section with basal lamina; B apical section with microvilli, arrows indicate ATER-system; C schematic overview; D pore canal in the cuticle, E recessed external surface over a pore canal; F inner surface of the gland cuticle, arrows indicate pore canals; G transection through the base of a setae with pore; bl = basal lamina, c = cuticle, fl = feltwork, g = Golgi apparatus, m = mitochondria, mv = microvilli, n = nucleus, pc = pore canal, rER = rough endoplasmatic reticulum, sc = setal canal, sj = septate junctions, v = vesicles.

Furthermore, the cells possess a well-developed cytoskeleton. Apically, the cells are closed by a microvilli border (Fig 2B). Within the individual microvilli, strands are visible, which are likely derived from the endoplasmic reticulum (ATER-system: Apical Tubular Extensions of the Rough Endoplasmic Reticulum). The microvilli border is connected to a narrow cavity, which is bounded by the cuticle on the opposite side. Numerous pore canals are visible in the gland cuticle (Fig 2B, 2D) and are associated with a characteristic felt-like network within the cavity (Fig 2B, 2D). Granular material is visible directly under and within the pore canals. The pore canals form an extensive network within the cuticular layers and give rise to lacunae within the cuticle (Fig 2D). No distinct excretory duct openings to the external surface or specialized pores penetrating the regular sternal gland cuticle or the setae surface were detected (Fig 3B-3F). Only a modified wax layer on the outer surface of the cuticle was visible (Fig 2E). The setae covering the outer surface of the cuticle in the range of the gland tissue (Fig 3D) exhibit a narrow basal opening that leads into the setal canal (Fig 3E-3G). These openings are clearly visible adjacent to the numerous cuticular pore canals (Fig 2F and 3F). The setal canal is initially hollow (Fig 1I, 2F and 3G) and becomes apically filled by a sponge-like structure (Fig 1I and 3H). The open lumen of the setal canal corresponds to the cavity between the microvilli fringe and the cuticle and is invaginated by the feltwork (Fig 2G). Within the setal cuticle, a pore canal system with lacunae was observed (Fig 3H).

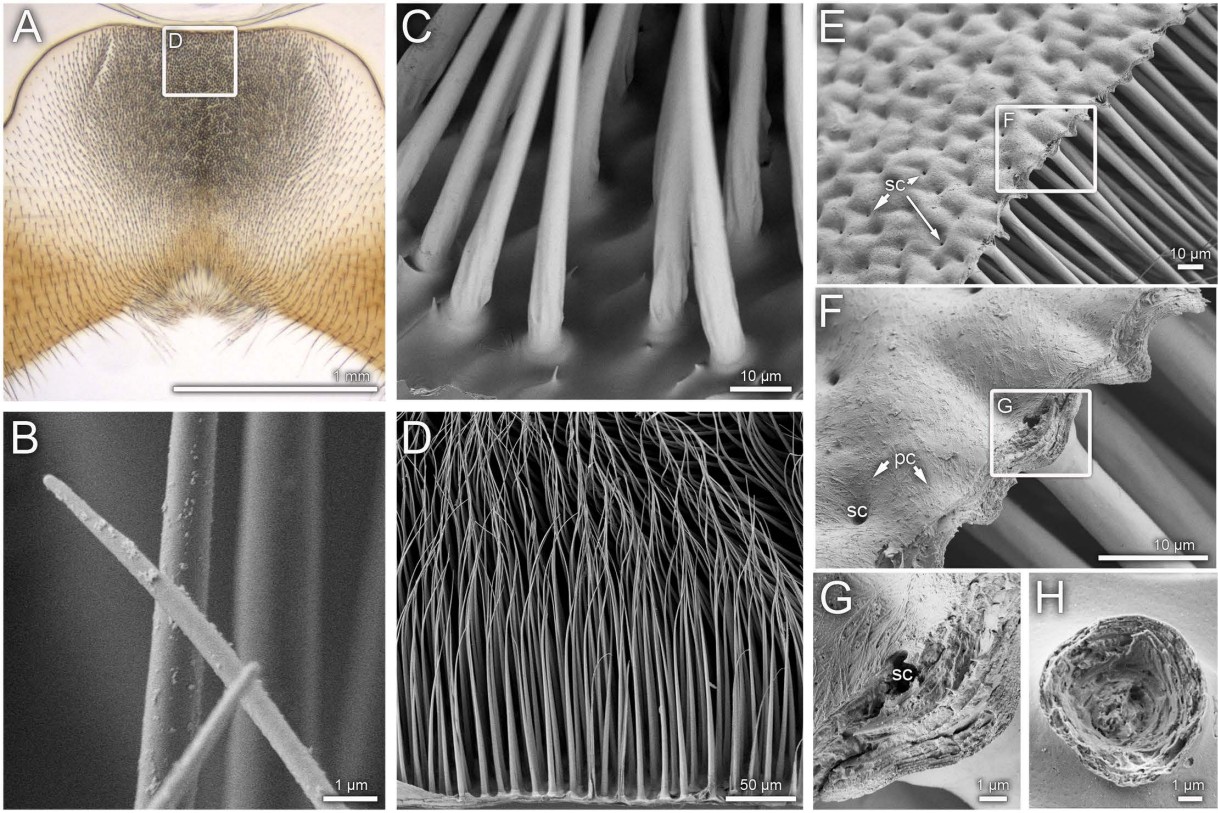

**Fig 3. Surface of the IV<sup>th</sup> sternite and setae.** A section of the IV<sup>th</sup> sternite (Whole Mount); B – D Setae covering the gland area; E, F rupture preparation of sternite showing internal surface of gland cuticle and adhering setae; G closeup of a setal canal; H rupture preparation of a setae close to the base; A, B, E – H: *O. bicornis*, C, D: *O. cornuta*.

## Effect of male-derived esters from sternal gland extracts

*Osmia cornuta* shows a much stronger SSB than *O. bicornis* based on the number of attempts to copulate before a male deserts from a once mounted male. Tattered newly emerged males were highly attractive to mate seeking conspecifics and were immediately mounted. The mounted male usually attempted to copulate several times before giving up and leaving. However, males became less attractive over time, resulting in a decrease in the duration of mountings and a decrease in the number of copulation (intromission) attempts. As tattered males became longer unattended over the course of trial, the number of approaches increased accordingly. The number of approaches was therefore not usable for this species. Tattered males were approached and mounted from 1 to 7 times (mean 2.11±1.45) prior to substance application and received thereby from 2 to 11 intromission attempts (mean 8.11±3.57) within the first 5 minutes. There was no difference in attractiveness between the males used for the different treatments and on different test days (ANOVA repeated measures, treatment×effect, F=2.041, p=0.157, pairwise comparisons all p>0.176; Fig 4). After application of the species-specific esters or the conspecific esters, the attractiveness of males was lower than that of solvent treated males (ANOVA repeated measures, treatment×effect, F=5.967, p=0.010, pairwise comparisons *cornuta* esters p=0.004, *bicornis* ester p=0,044; Fig 4).

Tattered *O. bicornis* males were attractive to mate seeking conspecifics but lost attractiveness much faster than *O. cornuta* males. Newly emerged males were mounted on the first approach and the mounting male attempted to copulate with the test male once or, in rare cases, twice, but departed soon. After a few minutes, tattered males were still approached

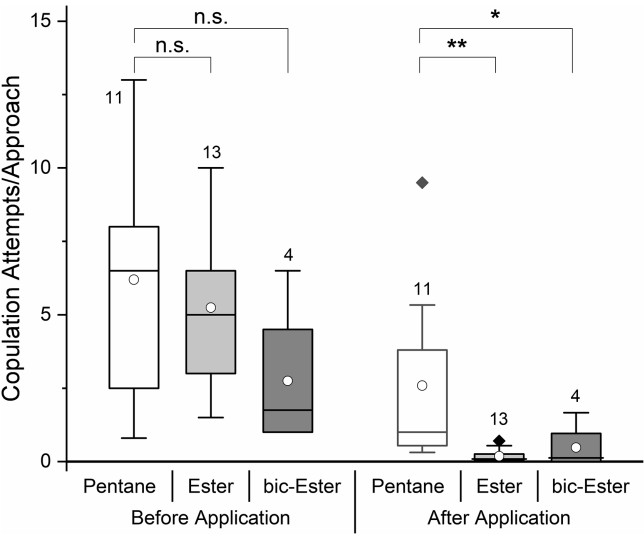

**Fig 4. Attractiveness of newly emerged *O. cornuta* males before and after application of sternal gland extracts or solvents.** Box plots indicate 5, 25, 50, 75, and 95 percentiles, outlying data points appear as diamonds, circles refer to the mean, numbers specify the number of trials; bic = *O. bicornis*, note that the alien extracts were tested at the end of the male emergence period with reduced SSB.

and mounted, but no intromission attempts were made. In addition, tattered males – especially those treated with esters – were still approached by males, but the males only made brief physical contact without mounting, or even turned away shortly before making contact. These approaches were classified as "inspections". Therefore, it was not possible to monitor the attractiveness of the males prior to the application of a substance. Instead, substances were applied immediately after tattering the male and differences in attractiveness between differently treated males had to be analysed. Due to the weaker SSB in this species, two parameters were chosen to assess attractiveness: The proportion of physical contacts made during all detectable approaches, including inspections, and the proportion of copulation (intromission) attempts of all cases in which the approaching male landed on the back of the tattered male (mounted it).

Tattered newly emerged males were approached 13.66±5.16 times within the observation period, depending on the trial and slightly on the treatment (ANOVA: trial: $F_{5, 53}$=12.622, p=0.003, treatment: $F_{3, 53}$=4.495, p=0.047, treatment×trial: $F_{7, 53}$=1.483, p=0.199). However, there was only one single approach with all the males treated with the *O. cornuta* esters (n=6) and this treatment was therefore excluded from the previous analysis. Overall, males were physically contacted by mate seeking conspecifics 0.93±0.09 times per approach. Application of sternal gland extracts reduced this contact frequency, whereas the application of the solvent pentane had no effect (univariate ANOVA, $F_{15, 57}$=38.162, p<0.01; trial: $F_{5, 57}$=3.056, p=0.020, treatment: $F_{3, 57}$=129.531, p<0.001, post-hoc Scheffé test: untreated×pentane p=0.153, all others p<0.001; Fig 5).

Males that mounted a tattered test male did not always attempt to initiate copulation. Even in untreated males, only 0.39±0.17 intromission attempts per mounting were observed. This rate was not influenced by pentane, but decreased with the application of the sternal gland extract, independent of the test day (ANOVA, trial: $F_{5, 57}$=0.809, p=0.578; treatment: $F_{3, 57}$=9.375, p=0.007; treatment×trial: $F_{7, 57}$=1.520, p=0.188; post-hoc Scheffé test: untreated×pentane p=0.994, all others p<0.001; Fig 6).

## Discussion

### Structure of the sternal gland and mode of volatile release

Males of both mason bee species produce sex-specific esters in homologous sternal glands, which are only present in males [10]. These glands are located on the IV$^{th}$ and V$^{th}$ sternite. Both sternites are shielded by the enlarged III$^{rd}$ sternite.

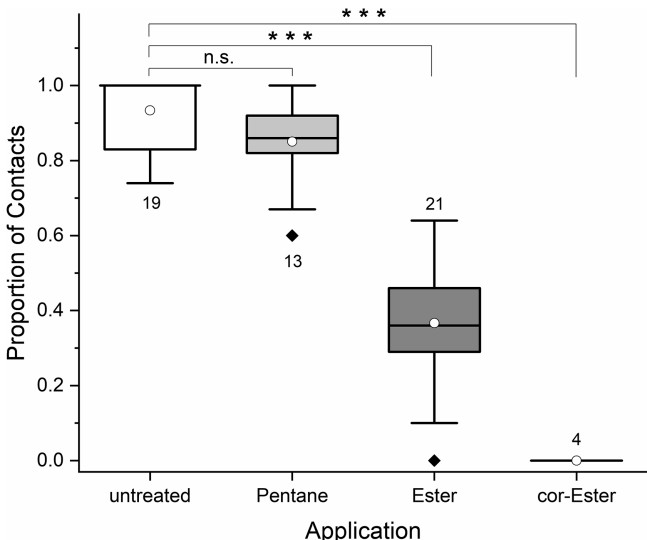

**Fig 5. Attractiveness of newly emerged *O. bicornis* males untreated, treated with sternal gland extracts or solvents, expressed as proportion of physical contacts per approach.** Box plots indicate 5, 25, 50, 75, and 95 percentiles, outlying data points appear as diamonds, circles refer to the mean, numbers specify the number of trials; cor = *O. cornuta*.

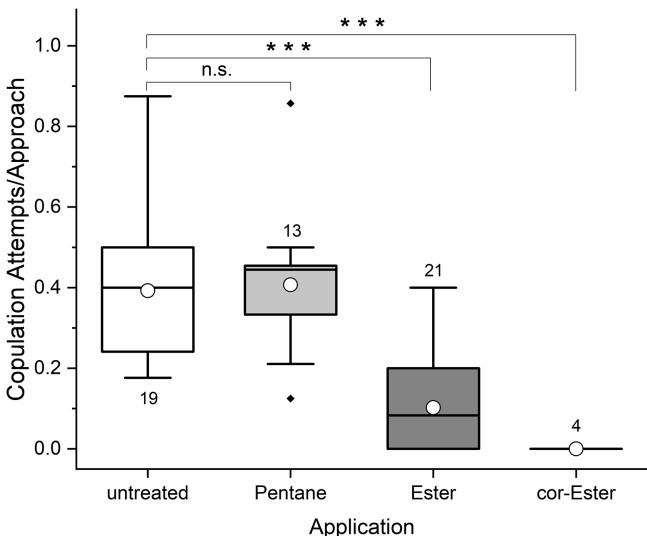

**Fig 6. Attractiveness of newly emerged *O. bicornis* males untreated, treated with sternal gland extracts or solvents expressed as copulation attempts per mount.** Box plots indicate 5, 25, 50, 75, and 95 percentiles, outlying data points appear as diamonds, circles refer to the mean, numbers specify the number of trials; cor = *O. cornuta*.

The sternal glands consist of secretory cells and an adhering cuticle, as it is typical of epidermal glands in general [27]. The gland cells exhibit the typical morphology of class 1 epithelial gland cells [12,34,35], the anatomical subtype without reservoir [36].

The secretion produced by the gland cells accumulates in a subcuticular reservoir, a small interspace between the gland cells and the cuticle. The cuticle above the cells is largely unmodified and lacks specific ducts for translocation of

the secretions to the outer surface of the sternites. Instead, secretions reach the exterior via expanded pore canals, a diffuse lacunar system, and epicuticular filaments. This is a common feature of class 1 epithelial gland cells [34,35] and has been observed, for example, in *Xylocopa micans* [37]. Although the canals do not open to the outside, shallow depressions in the wax layer may facilitate the penetration of pheromones to the surface.

Notably, the absence of pores and the restriction to class 1 gland cells is unusual for a sternal gland. By contrast, the sternal glands of other insects such as termites [16], ants [29], polistine wasps [20,23] and hornets [21] consist of a mixture of gland cell types with openings to extrude gland secretions. These insects often press the opening of an external reservoir against the subsurface to lay tracks (e.g., ants and termites), or use sternal brushes formed by setae to apply secretions to an object (wasps). In the case of the two mason bee species, however, pores are lacking. Thus, no liquid secretions can accumulate outside the cuticle [27]. The setae brush and hair bristle are covered by the III$^{rd}$ sternite, which prevents them from streaking secretions directly anyway.

The setae that cover the surface of the sternite over the gland cells have a sponge-like internal structure. Similar to the sternite cuticle covering the gland tissue, pore canals with lacunes penetrate into the cuticle without opening to the outside. The cavities within the setae connect to the space between the gland cells and the cuticle and are invaded by the feltwork at the base. It is likely that the sponge-like internal space is also filled with gland cell secretions. Consequently, the setae greatly increase the cuticular surface area and facilitate the release of pheromones. A surface extension of gland cuticle by setae or hairs is widely assumed to occur in various gland functions and insect taxa [27,38] and has been proven in *Helicoverpa zea* [39]. However, the mechanism of substance release in the mason bees remains unclear. Forced excretion by changing the haemolymph pressure [40] is unlikely due to the lack of pores. Exocrine glands that lack of pores and external reservoirs for secretions to accumulate following translocation are characterised by the uncontrolled, slow, and continuous release of secretions [27]. This pattern is typical of involuntary physiological functions.

Although the volatile compounds produced by the sternal gland of the male mason bees, such as the carboxylic esters, evaporate at a rather continuous rate from the gland's cuticle and setae surfaces into the gaseous phase, higher concentrations can be expected to accumulate in the small air volume of the caverns enclosed by the elongated III$^{rd}$ sternite. A somewhat functional similar structure of cavities lined up with setae, the hypertrophied dorsal mesosomal gland, is found in in four subgenera of carpenter bees (*Xylocopa*) [38]. This thoracic gland functions as both a secretory tissue and a large storage vessel from which the sex pheromone blend is released [37,38]. In *Osmia*, the enrichment of volatiles in a cavern enables a controlled discharge of pheromones through a post-release behaviour [27]. Higher concentrated pheromone puffs can be released through the ventilation of the enrichment vessel via the voluntary movement [27,41] of the elongated covering sternite and abdomen, respectively. Conspicuous movements of the abdomen, involving contractions and straightening movements upwards occur during the courtship and post-copulatory displays [32]. Moreover, it cannot be ruled out with certainty that the hair bristle at the end of the IV$^{th}$ sternite could be used to transfer secretions enriched in the hair's waxy layer to the hind legs. In addition to the continuous leakage of pheromones from the gland cavern this would distribute substances across the body surface during self-grooming.

## Pheromone effect of glandular secretions

The operational sex ratio is extremely imbalanced in both *O. cornuta* and *O. bicornis* due to female monandry and the pronounced male-biased sex ratio [4,5,42]. Consequently, males face strong competition to be the first to spot a virgin female [3,43]. Because females are receptive upon emergence, males are evolutionarily tuned to the CHC bouquet of overwintered cocooned bees, which is still present in newly emerged individuals [10]. A nuptial CHC bouquet is a reliable cue as the females of both *Osmia* species alter their CHC bouquet within a few days of emergence, coinciding with a decrease in mating readiness [8,32]. Females of both mason bee species do not release additional volatile sex pheromones [8]. Due to the low volatility of CHCs, males seeking a mate cannot identify an object as a conspecific

virgin female from a distance. They need to make physical contact with their antennae to sense the CHCs. Males are often seen touching, or even briefly grasping, mature females at rendezvous sites such as flowers or nests. Male-male encounters are also common (personal observations) and males run the risk of mistakenly grabbing a male or to being caught by a competitor.

To avoid being mistakenly grasped by a competitor, the males of both mason bee species enrich their CHC bouquet with large quantities of species-specific esters, which are more volatile than CHC compounds. These esters do indeed have a deterrent effect on other males. Topical application of male-specific esters significantly reduces the attractiveness of tethered males and SSB. The number of physical contacts made by males seeking mates, as well as the number of intromission attempts, declined or ceased altogether. This is consistent with the avoidance of males observed both in dummies and in females treated with 7-C16:1-EE by *O. bicornis* males [9,11]. However, applying the ester does not remove the virginity signal encoded by the CHC bouquet. This may explain why searching males still approach, and in rare cases even mount, ester-treated males. Additionally, virgin females that have only just been mounted by a male can still be detected by males seeking a mate at close quarters. This can lead to the formation of "mating balls" and piles with several males stacked on top of one another [4,5]. However, the attractiveness of couples decreases within a few minutes, with fewer and fewer males approaching the couple (personal observations). A similar effect was also observed in our study. Tethered males become less attractive over time. Patrolling males seem to learn the appearance of couples or tethered males and/or their position, and only inspect them, rather than mounting or landing. This was more obvious in *O. bicornis* than in *O. cornuta*.

The carboxylic esters used by *O. bicornis* and *O. cornuta* are chemically very similar. Therefore, it is not surprising that esters from the other species reduced the attractiveness of a newly emerged male to conspecifics. This effect was more pronounced in *O. bicornis* than in *O. cornuta*. The difference may be due to the complexity of the sternal gland compounds or to the generally higher probing activity in *O. cornuta*. The olfactory channel for carboxylic esters is obviously not narrowly tuned to the species-specific esters. The interspecific repellent effect may be due to the similarity of the substances that stimulate the same general olfactory receptors.

The carboxylic esters used by the males of the two mason bee species studied here are more volatile than CHC compounds. These esters act as male-deterrent pheromones or "abstinons" [44,45], and serve as an olfactory flag, announcing the sex of the male from a distance. Such a signal is advantageous for both the sender and the recipient. Males avoid vain contacts and captures, and do not waste time by courting competitors. However, we cannot rule out a feasible parsimony of the esters. Through the release of pheromone puffs via movements of the abdomen, the carboxylic esters could also act as courtship-inhibitory pheromones or antiaphrodisiacs, facilitating undisturbed mating [32].

Given that these ester-based pheromones are produced in specialized sternal glands, examining these glands across related species is crucial for understanding not only the diversity of chemical communication but also its evolutionary significance within the Osmiini tribe. Sternal glands are present in the males of various *Osmia* species and show considerable variation in their location and structure, particularly with regard to the presence of hair brushes or bristles (based on personal observations on a sample). Yet, the precise chemical composition and function of their secretions remain unknown for many species. Comparative analyses of gland morphology and secretory chemistry across species could therefore shed light on how these communication systems have diversified and adapted in response to selective pressures, thereby offering deeper insights into the evolutionary trajectories of mating strategies within the Osmiini.

## Supporting information

**S1. Striking image.**
(TIF)

## Acknowledgments

We are grateful to Joachim Händel (Central Repository for Natural Science Collections, Entomological collection) for assistance in direct-light micro photography. We thank Frank Syrowatka (Interdisciplinary Center of Materials Science, Electron microscopy lab) for help with scanning electron microscopy. We acknowledge the financial support of the Open Access Publication Fund of the Martin-Luther-University Halle-Wittenberg.

## Author contributions

**Conceptualization:** Karsten Seidelmann.

**Investigation:** Stephanie Krüger, Karsten Seidelmann.

**Resources:** Stephanie Krüger, Karsten Seidelmann.

**Visualization:** Stephanie Krüger.

**Writing – original draft:** Karsten Seidelmann.

**Writing – review & editing:** Stephanie Krüger, Karsten Seidelmann.

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
