## [Decision Letter · Decision Letter 0]

9 Sep 2025

PONE-D-25-42537Morphology, ultrastructure and function of the sternal gland in two mason bee species (Osmia bicornis and O. cornuta).PLOS ONE

Dear Dr. Seidelmann,

Thank you for submitting your manuscript to PLOS ONE. After careful consideration, we feel that it has merit but does not fully meet PLOS ONE’s publication criteria as it currently stands. Therefore, we invite you to submit a revised version of the manuscript that addresses the points raised during the review process.

Both reviewers have provided a number of constructive comments, to which we would ask you to respond briefly.

We look forward to receiving your revised manuscript.

Kind regards,

Wolfgang Blenau

Academic Editor

PLOS ONE

Journal Requirements:

3. Please note that your Data Availability Statement is currently missing [the repository name and/or the DOI/accession number of each dataset OR a direct link to access each database]. If your manuscript is accepted for publication, you will be asked to provide these details on a very short timeline. We therefore suggest that you provide this information now, though we will not hold up the peer review process if you are unable.

Reviewers' comments:

Reviewer's Responses to Questions

**Comments to the Author**

1. Is the manuscript technically sound, and do the data support the conclusions?

Reviewer #1: Yes

Reviewer #2: Yes

2. Has the statistical analysis been performed appropriately and rigorously? 

Reviewer #1: Yes

Reviewer #2: Yes

3. Have the authors made all data underlying the findings in their manuscript fully available?

Reviewer #1: Yes

Reviewer #2: Yes

4. Is the manuscript presented in an intelligible fashion and written in standard English?

Reviewer #1: Yes

Reviewer #2: Yes

5. Review Comments to the Author

Reviewer #1: The paper provides a nice description on the structure and potential function of the sternal gland in two species of Osmia. The protocols follow standards for the area and most of the analysis is well developed. However, the structure of the discussion requires change. Authors need to focus on their results and the relationship between these and literature, Latter, they can describe the higher implications of their study. The current version starts with statements about other studies that confuse the goal of their study. A stronmger closing argument root in their results is desirable.

Several minor comments are provided below.

1 Morphology and histology

2 Esthers as deterrents of harassment extending the role of their sex odors

3 Interspecific signaling function

Line 58. I suggest using homologous rather than analogous gland morphology as it is unlikely to be a convergence for these congeneric species and in such way the word usage is more rigorous.

Line 79. Please check typing for “signalling”

Lines 100-102. It is not clear the role of tissue maceration in the light and TEM microscopy section.

Line 136. Just for precision, authors used plastic petri dishes, right?

Please describe how do you test the effect of manipulation on males, how did you control cross contamination after new tested males are used for the study.

Line 159. Please check use of “ore” here

Lines 162. Please indicate the time frame considered for recording data.

As a general comment on morphologic terminology, in Apocrita the region after the constriction is named metasoma, not abdomen, as the propodeum is the first segment of the abdomen that, along the evolution of the group, is now strongly attached to the thorax. In some groups such as ants, this region is named gaster. Abdomen is more appropriate for other insects.

Lines 192 and 203. Usually figures are cited in order, thus, figure 2D should be after 2A, and 3A before than 3B.

Line 205. Authors described a basal opening and cite figure 3D, however, the opening is not obvious there.

Legend figure 4. Please provide the n of the trials.

Lines 190-197. The description is detailed but the figures and lettering in them are not very clear. Please improve visibility of the lettering on the EM images

Line 195. The lack of consistency in naming sternal glands in social wasps is largely discussed in 30 as this insect group holds several types of secretory cells in the same sternite.

Lines 298-299. This type of structural statements and a summary of results is desirable at the start of the discussion. It is a bit odd that authors started with comments on other papers rather than their own.

Line 305. Please check the bee name, it is Xylocopa. Finite may be replaced by fine or shallow depressions.

Line 312-313. May the authors explain better why the lack of pores imply absence of accumulation of the secretion?. It is possible that the secreted product permeates the insect wall and accumulate outside?

Lines 331-332. Please check style and grammar, there are several ideas but these appear unconnected or incomplete.

Lines 332, 345. Osmia should be in italics

The discussion may benefit of subtitles dealing with the two main topics treated.

Lines 365-367. Authors seems to suggest that visual cues and individual recognition is possible by the patrolling males. Please check and if so, provide support for this statement.

Lines 375-376. Here authors claim that emergence periods do not overlap and thus, chemical differentiation between species is not under pressure to occur, however, in the introduction (lines 78-79), they stated that interspecific signaling is a possibility due sympatry and thus, set an objective to test this idea. They may elaborate a different structure for this topic because the lack of overlapping at emergence is a strong reason, already available to suspect that interspecific signaling is not an issue.

Lines 384-386. Describing multiple potential explanations without further development or strong evidence is unnecessary and leaves the paper without a significant closing idea. The later is missing in the discussion, what are the most significant implications of the study? Where to go from here?

Reviewer #2: Congratulations on a fine study and paper. Results for Osmia bicornis and O. cornuta.

The study of exocrine glands in solitary (non-social) bees are rare. You have produced excellent histological data that convinces me that these sternal glands are real and have the sexual advertisement functions that you claim.

Its fascinating that the carboxylic esters from the sternal gland cells act as an anti-aphrodisiac (i.e. have a deterrent effect) and indicate the sex of the male from a short distance away.

Line 35. Yes, but male Osmia have to spend much of their time also finding flowers and drinking nectar to maintain a positive energy balance during their searches for virgin female bees as potential mates, and expending energy during fights with competing males. Yes, scramble competition polygyny as often described by John Alcock.

Line 41 should read "pouncing" on any object....

Male bees are tuned to the females' CHC bouquet, which is modified as the females age. You correctly state that newly emerged males can be confused (their cuticular hydrocarbons) with females. These males begin releasing large amounts of species-specific carboxylic acid esters within 3 days of emergence from their cocoons.

Your methods for fixation and histological prep. and examination are excellent. Ditto for SEM methods.

Lines 127 - 130. Please give me a bit more information about how males were presented to other males.

*These were tethered by a thin thread (made of ____?) at the petiole? You also made behavioral observations in the field as well as the laboratory, and these results agree.

Line 138. How were pentane extracts of sternal glands applied to male bees? How many microliters per bee and how long did you continue observations following topical pentane applications? That males were only tested once and then released is a good regimen.

Line 145. I see a mention of 5 minutes.

Your histological descriptions of the sternal glands, and associated photos, are excellent.

Histology of sternal gland cells. That the surfaces of these sternites are not more modified (just dented and with setae) is interesting.

I have no issues with your statistical measures of the SSB approaches etc, or stats in general. I see no reason for a statistician to further evaluate this paper. Use of ANOVA repeated measures test makes sense to me.

Figures 4, 5, and 6 are excellent. They both clearly show the results. I see no reason to make changes in these three figures.

Your histological results will likely lead other researchers to look for similar "hidden" sternal glands in other bees in the Megachilidae and elsewhere. There must be similar glands in other bees that we are not aware of, and yet are vitally important for an understanding of their courtship and copulatory behaviors.

Line 308. You correctly mention how different the sternal glands in your Osmia bees are from the sternal glands of other insects (e.g. termites, ants, Polistes wasps).

*The mechanism of "pheromone" release in Osmia remains unclear. I agree that it is likely a function of the uncontrolled, slow, and continuous release of these secretions. Perhaps discussions with a biophysicist interested in fluid dynamics would be appropriate, but not necessary for the publication of this article.

Lines 329 to 335. Yes, you correctly state that some dorsal mesosomal glands in Xylocopa bees are lined with many thousands of short still setae projecting into the lumen of their male glands. I have been part of such studies in Arizona. We do not fully understand the role of these setae in the release of pheromones from these massive glands. I am not sure that the role of setae for chemical dispersion in Xylocopa vs. Osmia are the same. I could be wrong.

Perhaps as you state, the Osmia chemical dispersion into the boundary layer air and beyond can be aided in the form of discrete puffs when the abdomen is ventilated to bring oxygen into the tracheae.

Line 378. I like the description of these anti-aphrodisiac chemicals as abstinons. A great word.

Olfactory flags. Male bees avoid vain captures and contacts. They do not waste time and energy by courting competitors.

**** One potential issue. I don't seem to have found the figure captions for Figures 1 through 6.

I'm sure they are fine, I just don't see them.

All in all, a wonderful paper, I can't wait to see it published in PLOS One.

6. PLOS authors have the option to publish the peer review history of their article (what does this mean?). If published, this will include your full peer review and any attached files.

Reviewer #1: **Yes: **Carlos E Sarmiento

Reviewer #2: **Yes: **Stephen L. Buchmann

---

## [Author Response · Author response to Decision Letter 1]

2 Oct 2025

There were no specific editor comments.

We addressed all the comments and suggestions from the reviewers in the 'Response to Reviewers' file. We have followed all the reviewers' suggestions point by point. There were no specific objections that needed to addresed here.

---

## [Editor Report · Decision Letter 1]

5 Oct 2025

Morphology, ultrastructure and function of the sternal gland in two mason bee species (Osmia bicornis and O. cornuta).

PONE-D-25-42537R1

Dear Dr. Seidelmann,

We’re pleased to inform you that your manuscript has been judged scientifically suitable for publication and will be formally accepted for publication once it meets all outstanding technical requirements.

Kind regards,

Wolfgang Blenau

Academic Editor

PLOS ONE

Additional Editor Comments (optional):

The authors have responded adequately to all points of criticism raised by the reviewers.
---

## [Editor Report · Acceptance letter]

PONE-D-25-42537R1

PLOS ONE

Dear Dr. Seidelmann,

I'm pleased to inform you that your manuscript has been deemed suitable for publication in PLOS ONE. Congratulations! Your manuscript is now being handed over to our production team.

Kind regards,

on behalf of

Dr. Wolfgang Blenau

Academic Editor

PLOS ONE